# Supramolecular polynuclear clusters sustained cubic hydrogen bonded frameworks with octahedral cages for reversible photochromism

Xiaojun Ding [1] ✉, Jing Chen[1] & Gang Ye [1] ✉

Developing supramolecular porous crystalline frameworks with tailor-made architectures from advanced secondary building units (SBUs) remains a pivotal challenge in reticular chemistry. Particularly for hydrogen-bonded organic frameworks (HOFs), construction of geometrical cavities through secondary units has been rarely achieved. Herein, a body-centered cubic HOF (TCA_NH$_4$) with octahedral cages was constructed by a C$_3$-symmetric building block and NH$_4^+$ node-assembled cluster (NH$_4$)$_4$(COOH)$_8$(H$_2$O)$_2$ that served as supramolecular secondary building units (SSBUs), akin to the polynuclear SBUs in reticular chemistry. Specifically, the octahedral cages could encapsulate four homogenous haloforms including CHCl$_3$, CHBr$_3$, and CHI$_3$ with truncated octahedron configuration. Crystallographic evidence revealed the cages served as spatially-confined nanoreactors, enabling fast, broadband photochromic effect associated with the reversible photo/thermal transformation between encapsulated CHI$_3$ and I$_2$. Overall, this work provides a strategy by shaping SSBUs to expand the framework topology of HOFs and a prototype of hydrogen-bonded nanoreactors to accommodate reversible photochromic reactions.

Construction of supramolecular porous structures with regular polyhedral architectures capturing guest molecules[1,2] offer opportunities for catalytic reaction[3,4], molecular recognition[5], and gas separation[6,7]. Most of the well-developed frameworks so far show open cavities with one or two-dimensional channels[8], lacking specific selectivity and dynamic sites for targeted objects[9,10]. Oppositely, synthesis of enclosed cavities with advanced symmetric three-dimensional cage architecture[11,12] or well-defined polyhedron[13] could implement efficient recognition and encapsulation of guest molecules[14,15]. At the same time, the isolated cavity could provide a potential platform and confined microenvironment for further physicochemical reactions[11,16].

For building sophisticated structures, nature gives representative examples to take advantage of secondary structures of proteins such as α-helix and β-sheet to assemble them into advanced living entities. There is no doubt that SBUs play important roles as nodes and clusters[17] in designing sophisticated substances[18]. The architectural and mechanical stability of metal-organic frameworks (MOFs) imparted by corresponding SBUs have given rise to unique framework chemistry. However, this concept has been rarely discussed in the design of HOFs with regular porosity, as HOFs are assembled by rigid building molecules through intermolecular hydrogen bonding[19–24]. Pristine hydrogen bonding interactions are too weak to stabilize rigid and directional polynuclear clusters[25–28]. The rational synthesis of porous HOFs with preorganized and highly symmetrical networks has been a long-term challenge[29]. Therefore, conventional hydrogen bonding motifs usually result in one-dimensional channels[30] instead of

[1]Collaborative Innovation Center of Advanced Nuclear Energy Technology, Institute of Nuclear and New Energy Technology, Tsinghua University, Beijing 100084, China. ✉e-mail: dingxj@mail.tsinghua.edu.cn; yegang@mail.tsinghua.edu.cn

hierarchical cavities[31], exhibiting poor adaptability to accommodate specific guest molecules[32,33]. The stacking manner of the building blocks defines the topology and porosity of HOFs[34–38]. Nevertheless, the weak nature of hydrogen bonds provides the possibility for the hybrid synthesis of hydrogen-bonded networks[39,40]. As a consequence, the employment of polyhedral clusters[41,42], diverse auxiliary interactions[43–45] such as charge-assisted hydrogen bonds[46–49] and multiple components[50–53] expands the opportunity to construct hydrogen-bonded networks based on more advanced, robust building units. At this point, $NH_4^+$, with a tetrahedral geometric structure stabilized by four equivalent N-H bonds and appropriate size ($r_{ionic} \approx 1.5$ Å), behaves in some ways like metal cations in crystal engineering. Cationic interaction of $NH_4^+$ could reliably contribute to electrostatic interactions through charge-assisted hydrogen bonds and potentially regulate the assembly of hydrogen-bonded networks. By donating ionic regulator in the assembly with organic building molecules, we envision that $NH_4^+$ offers tremendous opportunities for constructing advanced supramolecular architectures[54].

In this work, $NH_4^+$ node-assembled polynuclear clusters, akin to the SBUs in reticular chemistry, are exploited in HOF construction for the first time, enabling the formation of a body-centered cubic HOF (TCA_NH4) with distinctive molecular encapsulation, reversible and broadband photochromism, especially sensitive under 400 nm light irradiation. The polynuclear clusters $(NH_4)_4(COOH)_8(H_2O)_2$, denoted as SSBU-NH4−1, sustained by charge-assisted hydrogen bonds, are not as rigid and directional as the classic SBUs stabilized by metal-ligand coordination bonds, which are thus named as supramolecular secondary building units (SSBUs). The unit cell of TCA_NH4 contained an enclosed octahedral cage that could accommodate four haloforms such as $CHCl_3$, $CHBr_3$, and $CHI_3$. All of the encapsulated haloforms exhibited truncated octahedron configurations that suitably matched the enclosed octahedral cage. The distance of adjacent halogens within the encapsulated cavity decreased from Cl-Cl (4.513 Å), Br-Br (4.894 Å) to I-I (3.991 Å) along with decayed fluorescence intensity and shortened lifetime for the corresponding HOFs. Specifically, the host HOF with $CHI_3$ encapsulated in the confined octahedral cages, i.e., TCA_NH4@CHI3, showed fast photochromism (60 s) accompanied by fluorescence quenching outcome. Single crystal−single crystal transformation (SCSCT) analysis and electron paramagnetic resonance (EPR) revealed that $CHI_3$ was photolyzed to metastable $I_2$. More interestingly, such a photochromic reaction, occurring within the isolated octahedral cage of the host HOF, was reversible, and the color of TCA_NH4@CHI3 could be recovered through the heating-induced regeneration of $CHI_3$ from the metastable $I_2$.

## Results

Cubic single crystal of TCA_NH4 was obtained by evaporating the mixed solution of $C_3$-symmetric 4,4',4"-nitrilotribenzoic acid ($H_3$TCA)[55,56] and $NH_3 \cdot H_2O$ at room temperature (Supplementary Fig. 1). Blocky TCA_NH4 was also formed by $CH_2Cl_2$ diffusion in the mixed solution of $H_3$TCA and $NH_4Cl$ for weeks (Supplementary Fig. 2). SCXRD results showed $H_3$TCA crystallized in a body-centered cubic system and *I23* space group (Fig. 1a, Supplementary Table 1) with a cell length of 20.9737 Å. $NH_4^+$ served as nodes and bridged two neighboring $H_3$TCA molecules through neutral (2.509 Å) and charge-assisted N-H ( + )…O (-) (1.830 Å) hydrogen bonds (Fig. 1b, Supplementary Table 2). Water molecules strengthened the polynuclear clusters through O-H… O (2.069 Å) and N-H…O (2.478 Å) hydrogen bonds. Topologically, $H_3$TCA molecules can be considered as 3-connected nodes, one TCA connected three $NH_4^+$ through O-H…N hrdrogen bonds. Meanwhile, four $NH_4^+$ exactly arranged around two $H_2O$ molecule and connected eight $H_3$TCA molecules within the adjacent unit cell. Consequently, an octahedral cage with a size of about 12 Å was encompassed by $NH_4^+$ and $H_3$TCA molecules (Fig. 1c, Supplementary Fig. 3) and every unit cell contained an isolated cage (Fig. 1b, Supplementary Fig. 4).

As expected, $NH_4^+$ acted as the node and every four of them formed supramolecular clusters to connect neighboring cells (Fig. 1d, Supplementary Fig. 5).

Specifically, the above-obtained supramolecular polynuclear clusters $(NH_4)_4(COOH)_8(H_2O)_2$, with well-defined composition and structure sustained by hybrid hydrogen bonds, served as SSBUs in building the cubic network of TCA_NH4 with octahedral cavities. As shown in Supplementary Fig. 5, every $NH_4^+$ bridged two carboxy groups, and two $H_2O$ molecules further linked carboxy groups and $NH_4^+$ from two sides of the SSBUs. The SSBU-NH4-1 could be further considered as two oxygen (two $H_2O$) and four nitrogen (four $NH_4^+$) models. The simplified SSBU-NH4-1 interlaced orthogonally and connected with each other by O-H…O hydrogen bonds (Supplementary Figs. 5, 6). On the whole, the $NH_4^+$ node-assembled clusters not only connected each cell through six surfaces of cubic but also derived into SSBUs to construct advanced frameworks with enclosed octahedral cages.

The enclosed cage or cavity[57] appeared scarce since most HOFs possessed open channels[58–60] and pores[61–63] to transport guest molecules[64–69]. The pore structure of TCA_NH4 appeared an octahedral shape with solvent accessible volume of 14.9 %. However, the synthesized octahedral cavity of TCA_NH4 was isolated and surrounded by crystal water molecules, $NH_4^+$, and TCA molecules. $CO_2$ adsorption isotherm at 273 K of TCA_NH4 was conducted (Supplementary Fig. 7) and the low adsorption capacity indicated the pore can not be accessed by gas molecules at present. Interestingly, we found the void of TCA_NH4 could accommodate homogeneous haloforms including $CHCl_3$, $CHBr_3$, and $CHI_3$. The encapsulation was carried out in solvothermal synthesis at 70 °C in the presence of corresponding trihalomethane molecules. Cubic single crystals of TCA_NH4@CHCl3, TCA_NH4@CHBr3, and TCA_NH4@CHI3 were obtained within about 6 h (Supplementary Fig. 1). The encapsulation could also be achieved by $CH_2Cl_2$ diffusion in mixed solutions of $H_3$TCA and $NH_4Cl$ for weeks. As shown in Fig. 2a, SCXRD results found four homogeneous $CHX_3$ (X = Cl, Br, I) were assembled in the octahedral void within the unit cell. Trihalomethane formed C-H…N hydrogen bonds with four non-adjacent $H_3$TCA molecules (Fig. 2b). The angle of the three hydrogen bonds was all 180° but the distances slightly elongated with the increase of halogen radius (Supplementary Table 3). It was found that four trihalomethanes were precisely accommodated through truncated octahedron configuration (Fig. 2a, Supplementary Fig. 8), and each trihalomethane was parallel to one plane that contained one $H_3$TCA molecule. Additionally, $CHI_3$ formed C-H…I hydrogen bonds (2.817 Å) to enhance the encapsulation (Fig. 2c, Supplementary Table 4). SCXRD results of encapsulated crystals indicated that the encapsulation of trihalomethanes didn't change the crystal structure and the lattice constant and the volume of the unit cell was almost kept constant (Supplementary Table 5). But due to the inefficient and incompact encapsulation of $CHX_3$, the diffracted intensity of halogen atoms especially for iodine was slightly disordered, which implied that the octahedral cages were partly unoccupied by $CHX_3$ molecules. Still, the encapsulation of four trihalomethanes with truncated octahedron shape compatibly matched the octahedral cavity of the hydrogen-bonded framework. Powder X-ray diffraction (PXRD) analysis proved that after encapsulation of trihalomethane, the crystal structure of TCA_NH4 remained stable (Supplementary Fig. 9). Thermogravimetric (TG) results suggested that TCA_NH4, TCA_NH4@CHCl3, TCA_NH4@CHBr3, and TCA_NH4@CHI3 had similar thermal degradation behaviors (Supplementary Fig. 10) due to the encapsulation efficiency were too low and the weight percent of $CHX_3$ were also too low (<40 mg for encapsulation). It can be found that TCA_NH4 held moderate stability after immersing in organic solvent and water (Supplementary Fig. 11) compared with simulated PXRD result. The thermal and solvent stability of crystals indicated that SSBUs built by charged-assisted[70] supramolecular clusters stabilized TCA_NH4 and avoided

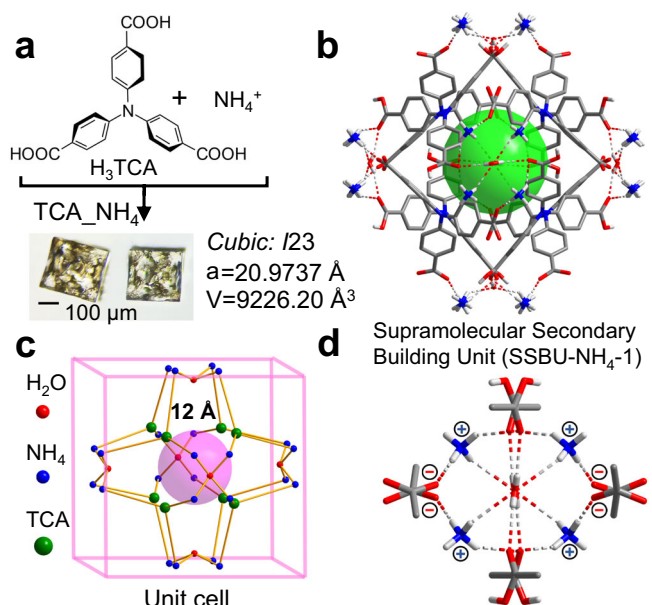

**Fig. 1 | Single-crystal structures of TCA_NH₄. a** Synthesis and shape of TCA_NH₄. **b** NH₄ charge-assisted hydrogen-bonded networks and isolated cavity (green color) within the unit cell. **c** Simplified topological network of TCA_NH₄ with TCA as a 3-connected nodes and octahedral cage (pink color). **d** SSBU·NH₄−1 of $(NH_4)_4(COOH)_8(H_2O)_2$ with concentrated polynuclear clusters and charge-assisted hydrogen bonds. Nonhydrogen bonding hydrogen atoms are omitted for clarity.

further phase change[71] and collapse of the framework. Taking advantage of the solution-processable benefit of HOFs[72], the encapsulated trihalomethane could be decapsulated by mild hydrolysis of hydrogen-bonded networks. After heating and dissolving in dimethyl sulfoxide (DMSO), nuclear magnetic resonance (NMR) results revealed that the encapsulated TCA_NH₄@CHCl₃, TCA_NH₄@CHBr₃, and TCA_NH₄@CHI₃ were observed to decompose and trihalomethane were observed to release (Fig. 2d). The encapsulation efficiency was roughly estimated by intensity of NMR spectra and all encapsulated samples were <10%, showing an incompact capture of trihalomethane.

With a closer examination, although the trihalomethane molecules were all encapsulated in truncated octahedrons, their configuration appeared slightly different. As shown in Fig. 3a, the distances of the nearest halogen atoms were classified as X1-X1, X1-X2, and X1-X3 (X = Cl, Br, I, Supplementary Fig. 12). Each classification included two directions. SCXRD results revealed that the distances of X1-X1 and X1-X2 were almost maintained after assembling CHX₃ (Fig. 3b). However, the distance of X1-X3 decreased from Cl1-Cl3 (4.513 Å) to Br1-Br3(4.894 Å), and then to I1-I3(3.99 Å). The shortened distances of adjacent halogen atoms provided the possibility for bonding reactions, especially for iodine. The color of TCA_NH₄ was changed after encapsulating haloforms, particularly for TCA_NH₄@CHBr₃ and TCA_NH₄@CHI₃. The color of crystals was resulted from the guest trihalomethane. In addition, the fluorescence emission spectrum of TCA_NH₄ showed no shift after assembling with CHCl₃ and CHBr₃ (Fig. 4a), but the emission spectrum red-shifted from 430 nm to 480 nm for TCA_NH₄@CHI₃ attributed to the absorbed energy transfer from H₃TCA to CHI₃. The absorbed energy transfer from H₃TCA to CHX₃ resulted red-shift of emission spectrum are largely determined by two reasons, one is the distances between H₃TCA and CHX₃, the other is the heavy atom effect-induced fluorescence quenching. The distances from H₃TCA to CHCl₃ and CHBr₃ are 2.805 Å and 3.007 Å, respectively (Supplementary Table 3), which are longer than the distance from H₃TCA to CHI₃ (2.244 Å). Therefore, the closer distance and heavy atom effect contributed to the absorbed energy transfer from H₃TCA to CHI₃ while the longer distances prevented the absorbed

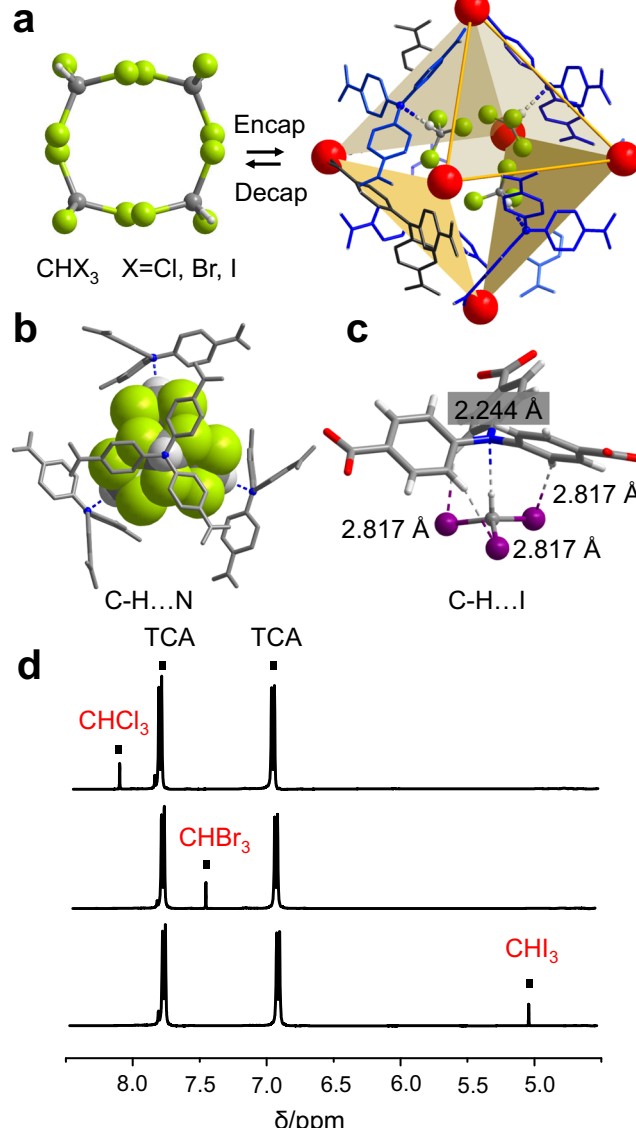

**Fig. 2 | Encapsulation and analysis of haloforms. a** Encapsulation and decapsulation of trihalomethane (CHX₃, X = Cl, Br, I) within octahedral cage. **b** C-H...N hydrogen bonds between trihalomethane and four noncontiguous TCA. **c** C-H...I hydrogen bonds for encapsulated CHI₃. **d** NMR spectra of TCA_NH₄@CHCl₃, TCA_NH₄@CHBr₃, and TCA_NH₄@CHI₃ in DMSO-d6. Nonhydrogen bonding hydrogen atoms are omitted for clarity.

energy transfer from H₃TCA to CHCl₃ and CHBr₃. Solid-state fluorescence reflection spectrum indicated that the fluorescence efficiency of TCA_NH₄ decreased (Supplementary Fig. 13) after encapsulating trihalomethane, because of heavy atom effect-induced fluorescence quenching. In the same way, the excitation spectrum of TCA_NH₄ also decreased after assembling haloforms, and the maximum excitation wavelength slightly blue-shifted from TCA_NH₄ at 410 nm to TCA_NH₄@CHI₃ at 380 nm (Supplementary Fig. 14) because of higher excitation energy for hydrogen-bonded networks of H₃TCA and trihalomethane. The lifetime of TCA_NH₄ was 2.09 ns (Fig. 4b, Supplementary Table 6) and it slightly decreased to 2.05 ns and 1.69 ns after assembling CHCl₃ and CHBr₃, respectively, following the quenching mechanism. But the lifetime increased to 2.43 ns after assembling CHI₃ maybe due to the energy exchange between excited species of H₃TCA and CHI₃ contributing to the delayed decay of the excited state. Solid-state absorption spectrum proved the structure of TCA_NH₄ did not vary obviously when compared with TCA_NH₄@CHCl₃,

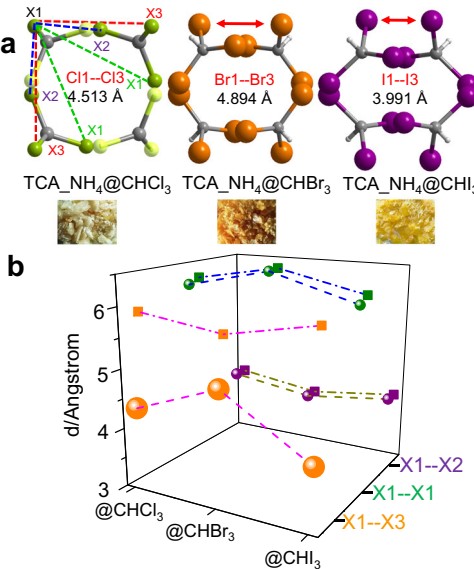

**Fig. 3 | Configuration of encapsulated haloforms. a** Color and configuration of encapsulated trihalomethane with defined distances between X1 and its neighboring atoms, the horizontal X1-X3 were decreased. **b** Distances of neighboring halogen atoms in encapsulated TCA_NH$_4$.

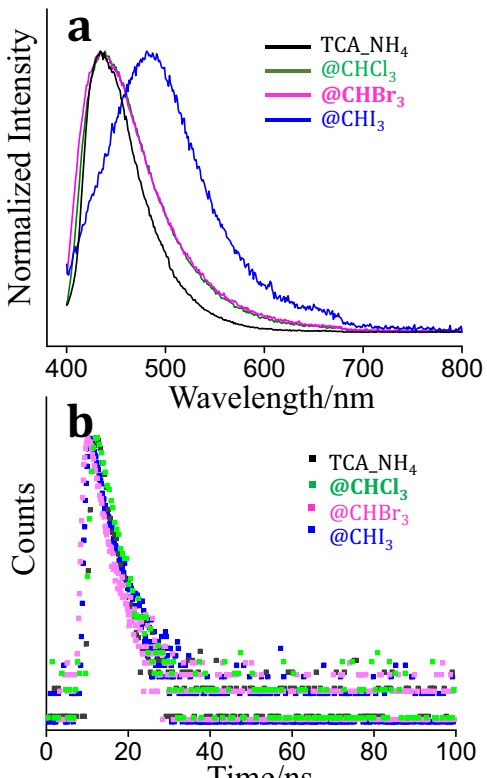

**Fig. 4 | Spectroscopic analysis of encapsulated haloforms. a** Normalized emission spectrum of TCA_NH$_4$, TCA_NH$_4$@CHCl$_3$, TCA_NH$_4$@CHBr$_3$, and TCA_NH$_4$@CHI$_3$ excited at 360 nm. **b** Decay profile of TCA_NH$_4$, TCA_NH$_4$@CHCl$_3$, TCA_NH$_4$@CHBr$_3$, and TCA_NH$_4$@CHI$_3$.

TCA_NH$_4$@CHBr$_3$, and TCA_NH$_4$@CHI$_3$ (Supplementary Fig. 15). The absorption peak appeared at 400 nm, indicating the crystals were more sensitive to ultraviolet light, especially for 400 nm of light. The above results indicated that the octahedral cavity within TCA_NH$_4$ precisely accommodated four trihalomethanes with matched

truncated octahedron shape. The color of assembled crystals changed by varied guest molecules. The fluorescence efficiency decreased after encapsulating haloforms. Particularly, the distances between adjacent halogen atoms were shortened especially for TCA_NH$_4$@CHI$_3$, which promoted energy transfer from excited H$_3$TCA to CHI$_3$ and prolonged lifetime and provided the possibility for bonding formation.

The encapsulation of photoactive CHI$_3$ inspired us to examine the photo-responsive behaviors of TCA_NH$_4$@CHI$_3$. Interestingly, TCA_NH$_4$@CHI$_3$ showed a 60-s fast photochromic phenomenon under irradiation of a UV-vis xenon lamp at 400 nm with a light power density of about 200 mW/cm² (Fig. 5a). SCSCT analysis revealed the distance of I1-I3 was further shortened to 2.822 Å of metastable I$_2$ approximately equivalent to I$_2$ (2.6 Å), and yellow TCA_NH$_4$@CHI$_3$ quickly transformed to brown TCA_NH$_4$@I$_2$. The solid-state absorption spectrum recorded the formation of metastable I$_2$ (Fig. 5b). The absorption peak of CHI$_3$ at 418 nm decreased and the absorption peak of metastable I$_2$ at 450 nm appeared after irradiation with UV-vis light for 5 min. Electron paramagnetic resonance (EPR) of TCA_NH$_4$@CHI$_3$ was conducted to examine the photolysis of iodoform. Disordered signals of TCA_NH$_4$@CHI$_3$ turned to specific radical signals (Fig. 5c) after in-situ illumination by a UV-vis light source with a light power density of about 200 mW/cm². The $g$ value at about 2.005 indicated the free electron resulted from the homolytic cleavage of the C-I bond[73] (Supplementary Fig. 16). Accordingly, TCA_NH$_4$@CHCl$_3$ and TCA_NH$_4$@CHBr$_3$ were also irradiated by a UV-vis light source to examine free radical signals. As shown in Supplementary Figs. 17 and 18, disordered signals indicated that no free radicals were generated or chemical bonds broken after irradiation. The absence of free radicals for TCA_NH$_4$@CHCl$_3$ and TCA_NH$_4$@CHBr$_3$ under illumination might be explained by two reasons. First, CHBr$_3$ and CHCl$_3$ are not intrinsically photosensitive. Meanwhile, the distances of Cl-Cl (4.513 Å) and Br-Br (4.894 Å) are longer than that of I-I (3.991 Å). As a result, only encapsulated photosensitive CHI$_3$ with a short distance of I-I bond showed unique reversible photo/thermal transformation. The SCXRD and EPR results proved that the photochromic reaction of TCA_NH$_4$@CHI$_3$ stemmed from photolysis of iodoform and the formation of metastable I$_2$. We consider the free radical generated within the enclosed cavity could further react with metastable I$_2$, on account of the isolated octahedral cavity providing a spatially-confined platform as nanoreactor that could stabilize free radicals and metastable I$_2$. As a result, after the photochromic reaction, brown TCA_NH$_4$@I$_2$ was immediately immersed in liquid paraffin to equably heat under a dark environment for an hour. As expected, the brown crystal returned to yellow and solid-state absorption spectrum results confirmed the recovery of iodoform (Fig. 5a, d). To examine the stability of the crystal after photo/thermal transformation, PXRD of TCA_NH$_4$@CHI$_3$ before and after irradiation under Xe lamp, as well as after thermal recovery, were conducted. As shown in Supplementary Fig. 19, the crystal structures maintained intact after reversible photo/thermal transformation as compared with the simulated PXRD pattern. The results indicated that HOFs encapsulated with CHI$_3$ could afford a stable photochromic material. Photochromic reaction and thermal recovery were repeated to investigate the reversible cycle of photo-responsive performance of TCA_NH$_4$@CHI$_3$. After repeating 5 rounds of irradiating and heating processes, TCA_NH$_4$@CHI$_3$ still maintained photochromic activity and the yellow crystal was still renewed (Supplementary Figs. 20, 21). The above results further proved that fast photochromic reaction and thermal recovery of encapsulated CHI$_3$ within an octahedral cage maintained a steady process to contribute as a new type of photochromic material.

For a photochromic process, usually, the dependence of light wavelength with a limited range restricts general applications in environment probing[74-76]. HOFs have shown great potential in luminescent fields[77]. Surprisingly, TCA_NH$_4$@CHI$_3$ not only turned color at 400 nm of light but also held a broad photosensitive scope ranging

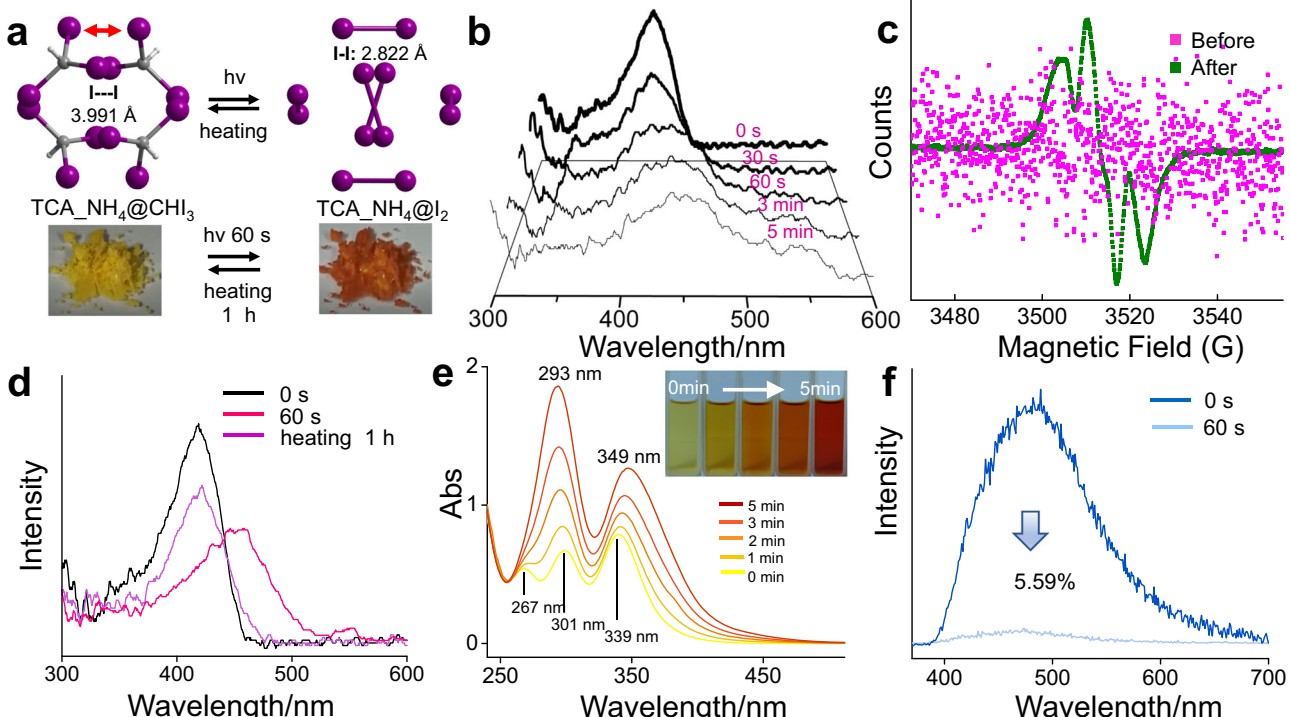

**Fig. 5 | Reversible photo/thermal transformation and characterization of encapsulated CHI₃. a** Photochromic reaction and thermal recovery of TCA_NH₄@CHI₃. **b** Solid-state absorption spectrum of TCA_NH₄@CHI₃ irradiated with a Xe lamp (400 nm, 200 mW/cm²). **c** EPR spectrum of TCA_NH₄@CHI₃ before and after irradiation under a UV-vis light source (200 mW/cm²). **d** Solid-state absorption spectrum of TCA_NH₄@CHI₃ after heating for an hour under dark environment. **e** UV-vis absorption spectrum of solid CHI₃ dissolved in ethanol (6 g/L) under irradiation with a Xe lamp for 5 min. **f** In-situ emission spectrum of TCA_NH₄@CHI₃ before and after irradiation with a Xe lamp for 60 s.

from X-ray, ultraviolet light, sunlight, and even low-energy light-emitting diode (8 mW/cm²). The varied light source and light intensity made TCA_NH₄@CHI₃ with different photosensitive efficiency ranging from 60 s to 5 min (Supplementary Table 7). The broad photosensitive scope and fast photochromic efficiency of TCA_NH₄@CHI₃ originated from the combination of multiple effects. Firstly, as shown by SCXRD in Fig. 3b, the shortened distance between adjacent I1-I3 in the enclosed octahedral cavity provided the possibility for a bond-formation reaction. Secondly, the decreased fluorescence efficiency of TCA_NH₄ after assembling trihalomethane promoted the transfer of the absorbed energy from H₃TCA to CHI₃, so that CHI₃ was energetic for potential physicochemical reactions. Thirdly, CHI₃ is intrinsically photosensitive. As shown in Fig. 5e, the pure CHI₃ powder was dissolved in ethanol (6 g/L) and irradiated under a xenon lamp with a light power density of about 200 mW/cm². The color of CHI₃ quickly turned brown within minutes (Inserted graph) and the UV-vis absorption spectrum proved the generation of I₂. The characteristic peaks of CHI₃ at 267 nm, 301 nm, and 339 nm decreased and characteristic peaks of I₂ at 293 nm and 349 nm appeared. Solid powder of CHI₃ also showed photosensitive reaction except that the color of the powder showed no visible change under 10 min irradiation with Xe lamp (Supplementary Fig. 22). Nevertheless, after irradiation with Xe lamp, both dissolved CHI₃ and solid powder of CHI₃ could not recover whether heating or standing due to the lack of an enclosed void for the restriction and stabilization of metastable products and free radicals. In addition, the fluorescence emission spectrum of TCA_NH₄@CHI₃ was markedly quenched to 5.59% after in-situ irradiation with Xe lamp for 60 s because of fluorescence quenching caused by metastable I₂ (Fig. 5f), and emission spectrum blue-shifted from 480 nm to 450 nm (Supplementary Fig. 23) that was close to TCA_NH₄@CHCl₃ and TCA_NH₄@CHBr₃ due to the break of energy transfer between H₃TCA and CHI₃. The excitation spectrum was also quenched after the

photochromic reaction (Supplementary Fig. 24) and the lifetime slightly decreased (Table S6, 2.43 ns vs 2.37 ns). The above results confirmed that the octahedral cavity within TCA_NH₄ offered unique nanoreactors for photochromic reaction and spatially-confined thermal recovery of CHI₃. The fast photochromic efficiency, broad photosensitivity, and unique fluorescence quenching of TCA_NH₄@CHI₃ showed great potential in sensing and environmental probing.

## Discussion

In summary, taking advantage of NH₄⁺ as nodes to bridge H₃TCA building blocks through charge-assisted hydrogen bonds, supramolecular polynuclear clusters (NH₄)₄(COOH)₈(H₂O)₂ were obtained and appropriately served as SSBU-NH₄-1 to construct a distinctive body-centered cubic hydrogen-bonded organic framework. The assembled TCA_NH₄ contained enclosed octahedral cages that precisely accommodated four homogenous trihalomethanes including CHCl₃, CHBr₃, and CHI₃. The truncated octahedron form of encapsulated trihalomethane compatibly matched the octahedral cavity of TCA_NH₄ through strong C-H...N hydrogen bonds. Besides, the fluorescence emission efficiency of TCA_NH₄ decreased after assembling different haloforms, promoting the transfer of the absorbed light energy to haloforms. The distances between adjacent halogen atoms within the octahedral cavity were shortened especially for CHI₃. As a result, TCA_NH₄@CHI₃ showed fast photochromic efficiency, broad photosensitivity, and unique fluorescence quenching behavior. The brown TCA_NH₄@I₂ could be recovered to TCA_NH₄@CHI₃ after heating due to the enclosed octahedral cages that provided spatially-confined nanoreactors to stabilize the free radical and metastable products. This study provides creative insights into the framework chemistry of HOFs and particularly an adaptable strategy for HOF construction from supramolecular polynuclear clusters sustained by charge-assisted H-bonds. The established methodology would expand the

framework topology of HOFs and accelerate the customized development of HOFs with predetermined architectures. Moreover, this study also provided a promising prototype of sensitive photochromic materials based on $CHI_3$ encapsulation within the isolated cavities of HOF nanoreactor platforms.

## Methods

### General remark

4,4',4"-nitrilotribenzoic acid ($H_3TCA$) and other reagents were purchased from Sinopharm Chemical Reagent Co., Ltd. in analytical grade. Powder X-ray diffraction data (PXRD) were performed by a Rigaku MiniFlex600 (40 kV, 15 mA) with a graphite-monochromatized Cu Kα radiation. Electron paramagnetic resonance (EPR) was conducted on a spectrometer (JEOL, JES FA-200). The data was collected with microwave field power of 0.7 mW. The modulation frequency was 100 kGHz and the microwave frequency was 9.7 GHz. Thermogravimetric analysis (TGA) was conducted on an SDT Q600 analyzer with a heating rate of 10 °C/min under $N_2$ (100 mL/min) atmosphere. $^1H$ NMR spectra was conducted in DMSO-d6 solution by JNM-ECZ400S (400 MHz) spectrometer.

### Synthesis of single crystals

**TCA_NH₄.** 120 mg $H_3TCA$ was dissolved in 0.8 mL N, N-Dimethylformamide (DMF) in a vial. 20 μL $NH_3H_2O$ was added and the mixed solution was slightly heated to get a clear solution. Then the solution was stood for evaporation and a cubic single crystal can be obtained after 3 weeks (Supplementary Fig. 1). Block single crystals also can be obtained by $CH_2Cl_2$ diffusion in a mixed solution of $H_3TCA$ and $NH_4Cl$ for 3 weeks.

**TCA_NH₄@CHCl₃.** 120 mg $H_3TCA$ was dissolved in 1 mL DMF in a vial. 40 μL $NH_3H_2O$ was added to get precipitation. 30 μL deionized water was added and the vial was slightly heated to get a clear solution. 30 μL $CHCl_3$ was added and the mixed solution was quietly evaporated at 70 °C. Cubic single crystals can be obtained within 6 h (Supplementary Fig. 1).

**TCA_NH₄@CHBr₃.** 120 mg $H_3TCA$ was dissolved in 0.8 mL DMF in a vial. 30 μL $NH_3H_2O$ was added to get precipitation. 40 μL deionized water was added and the vial was slightly heated to get a clear solution. 30 μL $CHBr_3$ was added and the mixed solution was quietly evaporated at 70 °C. Cubic single crystals can be obtained within 6 h (Supplementary Fig. 1).

**TCA_NH₄@CHI₃.** 200 mg $H_3TCA$ was dissolved in 0.8 mL DMF in a vial. 40 μL $NH_3H_2O$ was added to get precipitation. 80 μL deionized water was added and the vial was slightly heated to get a clear solution. 20 mg $CHI_3$ was added and the vial was heated to get a clear solution. Then the mixed solution was quietly evaporated at 66 °C under a dark environment. Yellow single crystals can be obtained within 6 h (Supplementary Fig. 1).

The encapsulation of $CHCl_3$, $CHBr_3$, and $CHI_3$ can also be obtained by $CH_2Cl_2$ diffusion in a mixed solution of $H_3TCA$, $NH_4Cl$, and trihalomethane for 3 weeks.

### Single-crystal X-ray diffraction analysis

Single crystal measurement was conducted on a Bruker APEX-II CCD with Cu Kα (λ = 1.54184 Å) X-ray sources. SADABS-2016/2(Bruker,2016/2) was used for absorption correction. The structure refinement was performed with Olex2 1.5 program. The structure was solved by ShelXT intrinsic phasing method and was refined by ShelXL least-squares techniques. All nonhydrogen atoms were refined with anisotropic displacement parameters. SQUEEZE function from PLATON program was taken to treat disordered solvent molecules in voids for TCA_NH₄.

The bond length of C-I bond was preset to be 2.0 Å in TCA_NH₄@I₂ using DFIX command to avoid disordered and free carbon atoms. Data collection and refinement details are listed in Table S1 and supplementary crystallographic CIF files have been deposited on Cambridge Crystallographic Data Centre (CCDC) with the number 2309944 for TCA_NH₄, 2309942 for TCA_NH₄@CHCl₃, 2309941 for TCA_NH₄@CHBr₃, 2309943 for TCA_NH₄@CHI₃, and 2309940 for TCA_NH₄@I₂.

### Solid-state spectroscopy measurement

Solid-state absorption spectroscopy was measured on a CRAIC 20/30 microspectrophotometer. A Xe lamp (90 W) was used as UV-vis and fluorescence source for measurement. For the UV-vis absorption test, parameters were set as Time1 = 57 ms: Objective = 15X: Aperture = $4 \times 4$ mm². For the fluorescence reflection test, 365 nm channel was used and parameters were set as Time1 = 1000 ms: Objective = 15X: Aperture = $4 \times 4$ mm².

### The fluorescence emission and absorption measurement

The data was collected on a FLS1000 spectrometer with Visible/red-PMT detector and Xe lamp (300 W) as the light source for measurement. The excitation wavelength was 360 nm (bandwidth: 1 nm) and the emission wavelength was 470 nm (bandwidth: 0.4 nm). A nF920 nanosecond flashlamp ($H_2$ padded in chamber) was used for lifetime measurement. The time range was <500 millisecond and channels were fixed to 1024, and stop condition was set to 1000 counts. The decay lifetime was fitted according to multiexponential function models. The matching rate of fitting results were estimated by $\chi^2$ (1.0-1.1, Table S6).

### Photochromic reaction and thermal recovery

The synthesized yellow TCA_NH₄@CHI₃ crystal was washed with dimethyl formamide (DMF) and acetone and then dried out. The yellow crystal was put into a vial and heated at 60 °C under a dark environment for 30 min to activate the crystal. Then the crystal was put under a Xe lamp (200 mW/cm², 400 nm) to be exposed for 1 min or directly exposed under intense sunlight for 1–5 min. The color of the crystal quickly turned to brown and then the crystal was immediately immersed in liquid paraffin to equably heat at 100 °C for an hour under dark surroundings. The color of the crystal recovered to yellow. The crystal was taken out and washed with organic solvent to remove liquid paraffin and then dried out. The recovered crystal was preserved under dark surroundings.

## Data availability

The data that support the findings of this study are available in the Supplementary Information. The X-ray crystallographic files for structures in this study have been deposited at the Cambridge Crystallographic Data Centre (CCDC), with numbers 2309940-2309944. These data can be obtained free of charge from The Cambridge Crystallographic Data Centre via www.ccdc.cam.ac.uk/getstructures. The source data generated in this study have been deposited in the Figshare database under accession code DOI: 10.6084/m9.figshare.25272415. Source data are provided with this paper.

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

## Acknowledgements

We acknowledge the National Natural Science Fund for Excellent Young Scholars (No. 21922604, G.Y.), the National Natural Science Foundation of China (No. 22376117, G.Y.; No. 22206105, X.D.), and Tsinghua University Initiative Scientific Research Program (G.Y.) for the financial support of this work.

## Author contributions

X.D. and G.Y. designed the experiment and synthesized crystalline materials. X.D. and G.Y. conducted the experiment and wrote the article. J.C. provided valuable suggestions for the study.

## Competing interests

The authors declare no competing interests.
