## [Peer Review File · Nature Communications]

REVIEWER COMMENTS

Reviewer #1 (Remarks to the Author):

Hydrogen bonded enhanced frameworks are a new field with not many examples described in the literature. It is understandable that this type of frameworks are difficult to design and therefore to synthesize. The work described in here can be considered relevant to the field, as it is trying to design cavities based on principles compatible with reticular chemistry, which itself would represent a step forward in the field. Additionally, the use of these designed cavities as catalytic cavities is also of interest. Therefore, I recommend its publication in Nat. Communications.

Some comments that I consider that should be addressed:

- Figure S10 should include the simulated PXRD for better comparison.
- Why TGA was not used to determine the amount of CHX encapsulated and compared with the NMR?
- In the part of reversible photo/thermal transformation of CHI₃, is there any precedent in the literature?
- One important property that the authors are not discussing is the permanent porosity or gas adsorption properties. Have the authors measured the gas adsorption?

Reviewer #2 (Remarks to the Author):

In this manuscript, the author prepared supramolecular polynuclear clusters NH₄⁺ as nodes to bridge H₃TCA building blocks, which served as SSBUs to construct a

distinctive body-centered cubic hydrogen-bonded organic framework. The octahedral cavity within TCA_NH₄ can accommodate four homogenous trihalomethanes and offer unique nanoreactors for reversible photochromic reaction and spatially-confined thermal recovery of CHI₃. Overall, the paper is mainly based on analysis of experimental results, and the results are interesting. I recommend a minor revision for this manuscript. Here are some questions for the authors.

1. In Page 3, Column right, the author mentioned "red-shifted from 430 nm to 480 nm for TCA_NH₄@CHI₃ attributed to the absorbed energy transfer from H₃TCA to CHI₃". Please give the underlying causes for the absence of the absorbed energy transfer from H₃TCA to CHBr₃ and CHCl₃.
2. The author should provide EPR spectrum with magnetic field and value g signals of TCA_NH₄@CHBr₃ before and after irradiation.
3. Powder X-ray diffraction of TCA_NH₄@CHI₃ before irradiation and after 238 irradiation under Xe lamp should be performed to confirm the intact crystal structure of TCA_NH₄.

4. There are also some grammatic mistakes. For instance, "H3TCA molecules can be considered as 3-connected nodes, it can be found NH4 + exactly arranged as the round to connect eight H3TCA molecules within the adjacent unit cell." in Page 2, Column left.

5. The ESI figures were mislabeled. For instance, (b) was omitted in "Figure S16. EPR spectrum with magnetic field (a) and value g signals of TCA_NH4@CHCl3"

Reviewer #3 (Remarks to the Author):

The authors reported hydrogen-bonded organic frameworks (HOFs) constructed from node-assembled cluster so-called supramolecular secondary building units (SSBUs). The HOFs possess octahedral cages, in which four molecules of haloforms (CHCl₃, CHBr₃, or CHI₃) can be accommodated. The haloforms included in structurally restricted voids in the HOFs reversibly undergo photolysis and thermal recombination, allowing photochromic behaviors. The manuscript seems to be well written. The photolysis of haloforms were well investigated by single X-ray and EPR analyses. However, the important idea in the manuscript, "supramolecular secondary building units" is not new, although the exact term have not been appeared. For example, the idea of SSBU has been proposed in 2007 by Tohnai and used for construction of crystalline framework materials possessing discrete and 1D-channeled void space. For example, see the following literature:

Angew. Chem., Int. Ed. 2007 46 2220.

ChemNanoMat 2023, 9, e202300248.

Small 2023, 19, 2301887, although the authors did not cite them.

Additionally, these kinds of materials possessing discrete voids capable of accommodating some guest species were intensively investigated mainly in 89-90s. These materials were called inclusion compounds. Such caged structures are also seen in some kinds of hydrates. Based on these backgrounds, the reviewer recommends to submitting more specific journals on crystal engineering, material science, or solid state chemistry.

● Point-by-point response to reviewers

Reviewer #1

Comments: Hydrogen bonded enhanced frameworks are a new field with not many examples described in the literature. It is understandable that this type of frameworks are difficult to design and therefore to synthesize. The work described in here can be considered relevant to the field, as it is trying to design cavities based on principles compatible with reticular chemistry, which itself would represent a step forward in the field. Additionally, the use of these designed cavities as catalytic cavities is also of interest. Therefore, I recommend its publication in Nat. Communications.

Some comments that I consider that should be addressed:

Response: We appreciate the reviewer for the positive comments. The recent development in the reticular chemistry inspires us to explore strategies adaptable for HOF synthesis. The design principle based on NH_4^+ node-assembled polynuclear clusters represents a small step forward. We have revised the manuscript based on the reviewer's comments as follows.

Comment 1: Figure S10 should include the simulated PXRD for better comparison.

Response 1: We have added the simulated PXRD in Supporting Information Figure S10 (revised as Supplementary Fig. 11) for comparing the stability of TCA_NH_4 in different solvents. Related discussions have been revised accordingly as follows:

It can be found that TCA_NH_4 held moderate stability after immersing in organic solvents and water (Supplementary Fig. 11) compared with the simulated PXRD result. (Page 3, right column, highlighted in yellow)

Supplementary Figure 11. PXRD of TCA_NH_4 after immersing in different solvents as compared with the simulated pattern.

Comment 2: Why TGA was not used to determine the amount of CHX encapsulated and compared with the NMR?

Response 2: Thanks for the constructive comment. For our samples, TGA is not a convenient tool to determine the amount of the encapsulated CHX₃, since the decapsulation occurs in the pyrolysis process between 200 °C to 300 °C, which overlaps with the decomposition or release of other species including crystal water, NH₄⁺, and the disordered solvents within the cavities. Thus, the encapsulation efficiency was evaluated by dissolving encapsulated TCA_NH₄@CHCl₃, TCA_NH₄@CHBr₃, and TCA_NH₄@CHI₃ in DMSO and the amount of CHX₃ was calculated based on the NMR spectra.

Comment 3: In the part of reversible photo/thermal transformation of CHI₃, is there any precedent in the literature?

Response 3: To the best of our knowledge, this is the first paper reporting the reversible photo/thermal transformation of CHI₃ encapsulated within spatially confined HOF cavities. Meanwhile, this work also presented the fast, broadband photochromic effect of encapsulated CHI₃ within the octahedral cages, making the material potential for environmental probing.

Comment 4: One important property that the authors are not discussing is the permanent porosity or gas adsorption properties. Have the authors measured the gas adsorption?

Response 4: Thanks for the constructive comment. The synthesized porous TCA_NH₄ contain isolated octahedral cavities that are surrounded by crystal water molecules, NH₄⁺, and TCA molecules. Therefore, the cavities cannot be accessed by gas molecules at present. We have conducted the CO₂ adsorption isotherm at 273 K of TCA_NH₄ in the revised manuscript. The results confirmed that CO₂ adsorption capacity was very low. Relevant discussions have been added in the revised manuscript as follows:

The pore structure of TCA_NH₄ appeared an octahedral shape with solvent accessible volume of 14.9%. However, the octahedral cavities of TCA_NH₄ were isolated and surrounded by crystal water molecules, NH₄⁺, and TCA molecules. CO₂ adsorption isotherm at 273 K of TCA_NH₄ was conducted (Supplementary Fig. 7) and the low adsorption capacity indicated the pores could not be accessed by gas molecules at present. (Page 2, right column, highlighted in yellow)

Supplementary Figure 7. CO₂ adsorption isotherm of TCA_NH₄ at 273 K (red adsorption, black desorption).

Reviewer #2

Comment: In this manuscript, the author prepared supramolecular polynuclear clusters NH₄⁺ as nodes to bridge H₃TCA building blocks, which served as SSBU to construct a distinctive body-centered cubic hydrogen-bonded organic framework. the octahedral cavity within TCA_NH₄ can accommodate four homogenous trihalomethanes and offer unique nanoreactors for reversible photochromic reaction and spatially-confined thermal recovery of CHI₃. Overall, the paper is mainly based on analysis of experimental results, and the results are interesting. I recommend a minor revision for this manuscript. Here are some questions for the authors.

Response: We appreciate the reviewer for the positive evaluation of this paper. We have revised the manuscript as follows.

Comment 1: In Page 3, Column right, the author mentioned "red-shifted from 430 nm to 480 nm for TCA_NH₄@CHI₃ attributed to the absorbed energy transfer from H₃TCA to CHI₃." Please give the underlying causes for the absence of the absorbed energy transfer from H₃TCA to CHBr₃ and CHCl₃.

Response 1: The red-shift of the emission spectrum of TCA_NH₄@CHI₃ associated with the absorbed energy transfer from H₃TCA to CHX₃ may be linked to two potential causes. One is the heavy atom effect-induced fluorescence quenching, and the other is

the difference in the distance between H₃TCA and CHX₃. Specifically, the distances from H₃TCA to CHCl₃ and CHBr₃ are 2.805 Å and 3.007 Å, respectively (Supplementary Table 3), which are significantly longer than that from H₃TCA to CHI₃ (2.244 Å). Therefore, the closer distance, combined with the heavy atom effect, contributed to the absorbed energy transfer from H₃TCA to CHI₃, while the longer distances prevented the absorbed energy transfer from H₃TCA to CHCl₃ and CHBr₃. Corresponding discussions have been included in the revised manuscript as follows:

In addition, the fluorescence emission spectrum of TCA_NH₄ showed no shift after assembling with CHCl₃ and CHBr₃ (Fig. 4a), but the emission spectrum red-shifted from 430 nm to 480 nm for TCA_NH₄@CHI₃ due to the absorbed energy transfer from H₃TCA to CHI₃. The red-shift of the emission spectrum associated with the absorbed energy transfer is attributed to two reasons. One is the heavy atom effect-induced fluorescence quenching, and the other is the difference in the distance between H₃TCA and CHX₃. The distances from H₃TCA to CHCl₃ and CHBr₃ are 2.805 Å and 3.007 Å, respectively (Supplementary Table 3), which are significantly longer than the distance from H₃TCA to CHI₃ (2.244 Å). Therefore, the heavy atom effect, combined with the closer intermolecular distance, contributed to the absorbed energy transfer from H₃TCA to CHI₃, while the longer distances prevented the absorbed energy transfer from H₃TCA to CHCl₃ and CHBr₃. (Page 3, right column, highlighted in yellow)

Supplementary Table 3. C-H...N hydrogen bond parameters of TCA_NH₄@CHCl₃, TCA_NH₄@CHBr₃, and TCA_NH₄@CHI₃.

	d/C-H...N	θ/C-H...N	Unit Cell Volume
TCA_NH ₄	--	--	9226.2 Å ³
TCA_NH ₄ @CHCl ₃	2.805 Å	180°	9221.2 Å ³
TCA_NH ₄ @CHBr ₃	3.007 Å	180°	9207.7 Å ³
TCA_NH ₄ @CHI ₃	2.244 Å	180°	9204.7 Å ³

Comment 2: The author should provide EPR spectrum with magnetic field and value g signals of TCA_NH₄@CHBr₃ before and after irradiation.

Response 2: Thanks for the constructive suggestion. We conducted EPR analysis of TCA_NH₄@CHBr₃ before and after irradiation and found the results were consistent with TCA_NH₄@CHCl₃. No free radical signal was detected before or after irradiation with corresponding light. These phenomena could be explained by two facts. First, CHBr₃ and CHCl₃ are not intrinsically photosensitive. Meanwhile, the distances of Cl-Cl (4.513 Å) and Br-Br (4.894 Å) are longer than that of I-I (3.991 Å). As a result, only

encapsulated photosensitive CHI_3 with a shorted distance of I-I showed unique reversible photo/thermal transformation. Related texts have been revised accordingly:

Accordingly, $\text{TCA_NH}_4@\text{CHCl}_3$ and $\text{TCA_NH}_4@\text{CHBr}_3$ were also irradiated by a UV-vis light source to examine free radical signals. As shown in Supplementary Fig. 17 and Supplementary Fig. 18, disordered signals indicated that no free radicals were generated or chemical bonds broken after irradiation. The absence of free radicals for $\text{TCA_NH}_4@\text{CHCl}_3$ and $\text{TCA_NH}_4@\text{CHBr}_3$ under illumination might be explained by two reasons. First, CHBr_3 and CHCl_3 are not intrinsically photosensitive. Meanwhile, the distances of Cl-Cl (4.513 Å) and Br-Br (4.894 Å) are longer than that of I-I (3.991 Å). As a result, only encapsulated photosensitive CHI_3 with a short distance of I-I bond showed unique reversible photo/thermal transformation. (Page 4, right column, highlighted in yellow)

Supplementary Figure 18. EPR spectrum with magnetic field (a) and value g signals of $\text{TCA_NH}_4@\text{CHBr}_3$ before and after irradiation with UV-vis lamp power density of 200 mW/cm^2 .

Comment 3: Powder X-ray diffraction of $\text{TCA_NH}_4@\text{CHI}_3$ before irradiation and after irradiation under Xe lamp should be performed to confirmed the intact crystal structure of TCA_NH_4 .

Response 3: Thanks for the constructive suggestion. We conducted PXRD of $\text{TCA_NH}_4@\text{CHI}_3$ before and after irradiation under a Xe lamp. We also performed the PXRD of $\text{TCA_NH}_4@\text{CHI}_3$ after photo/thermal transformation. As shown in Supplementary Fig. 19, the crystal structure maintained intact after reversible photo/thermal transformation compared with simulated PXRD. The results indicated that encapsulated CHI_3 within HOFs formed a stable photochromic material. Corresponding discussions have been included in the revised manuscript as follows:

To examine the stability of the crystal after photo/thermal transformation, PXRD of $\text{TCA_NH}_4@\text{CHI}_3$ before and after irradiation under Xe lamp, as well as after thermal recovery, were conducted. As shown in Supplementary Fig. 19, the crystal structures maintained intact after reversible photo/thermal transformation as compared with the simulated PXRD pattern. The results indicated that HOFs encapsulated with CHI_3 could afford a stable photochromic material. (Page 5, left column, highlighted in yellow)

Supplementary Figure 19. PXRD of $\text{TCA_NH}_4@\text{CHI}_3$ before and after irradiation under Xe lamp and after thermal recovery.

Comment 4: There are also some grammatic mistakes. For instance, “H₃TCA molecules can be considered as 3-connected nodes, it can be found NH₄⁺ exactly arranged as the round to connect eight H₃TCA molecules within the adjacent unit cell.” in Page 2, Column left.

Response 4: Thanks for the comment. The sentences have been revised as follow:

Topologically, H₃TCA molecules can be considered as 3-connected nodes. One TCA connected three NH₄⁺ through O-H...N hydrogen bonds. Meanwhile, four NH₄⁺ were exactly arranged around two H₂O molecules and connected eight H₃TCA molecules within the adjacent unit cells. (Page 2, left column, highlighted in yellow)

Comment 5: The ESI figures were mislabeled. For instance, (b) was omitted in “Figure S16. EPR spectrum with magnetic field (a) and value g signals of TCA_NH₄@CHCl₃”

Response 5: The caption of Figure S16 (revised Supplementary Fig. 17) has been corrected as shown below:

Supplementary Figure 17. EPR spectrum with the magnetic field (a) and value g signals (b) of TCA_NH₄@CHCl₃ before and after irradiation with UV-vis lamp power density of 200 mW/cm².

Reviewer: 3

Comment: The authors reported hydrogen-bonded organic frameworks (HOFs) constructed from node-assembled cluster so-called supramolecular secondary building units (SSBUs). The HOFs possess octahedral cages, in which four molecules of haloforms (CHCl₃, CHBr₃, or CHI₃) can be accommodated. The haloforms included in structurally restricted voids in the HOFs reversibly undergo photolysis and thermal recombination, allowing photochromic behaviors. The manuscript seems to be well written. The photolysis of haloforms were well investigated by single X-ray and EPR analyses. However, the important idea in the manuscript, “supramolecular secondary building units” is not new, although the exact term has not been appeared. For example, the idea of SSBUs has been proposed in 2007 by Tohnai and used for construction of crystalline framework materials possessing discrete and 1D channeled void space. For example, see the following literature: *Angew. Chem., Int. Ed.* 2007, 46, 2220. *ChemNanoMat* 2023, 9, e202300248. *Small* 2023, 19, 2301887, although the authors did not cite them.

Additionally, these kinds of materials possessing discrete voids capable of accommodating some guest species were intensively investigated mainly in 89-90s. These materials were called inclusion compounds. Such caged structures are also seen

in some kinds of hydrates. Based on these backgrounds, the reviewer recommends to submitting more specific journals on crystal engineering, material science, or solid-state chemistry.

Response: We appreciate the reviewer's professional and insightful comments, particularly the emphasis on relevant supramolecular concepts in previous contributions.

As we know, hydrogen-bonded (H-bonded) structures have been studied since half a century ago. Active researchers in this area include Duchamp and Marsh, Herbstein, Ermer and Wuest, *etc.* In the 1990s, the terms "tecton" and "synthon" made it clear for the construction of porous molecular crystals, and more effective synthetic strategies were then established for building porous H-bonded molecular crystals. Specifically, Tohnai developed sulfonic acids and amines charge-assisted H-bonding motifs. Sozzani studied the permanent porosity of crystals through laser-polarized ^{129}Xe NMR spectra. Hisaki studied C_3 - and C_6 -symmetric structures based on carboxylic acid dimer H-bonds. Although relevant works have been studied for years, it was not until 2011, when Chen re-investigated "tecton-1" and found its great potential in gas adsorption and separation, that the concept of "hydrogen-bonded organic frameworks" (HOFs) was proposed.

The classification of H-bonded porous structures as HOFs, reminiscent of the MOFs and COFs in reticular chemistry, has driven rapid progress in this area. However, due to the weak nature of H-bonds, predictable and adaptable synthetic strategies have not been established. We present in this study a body-centered cubic HOF with octahedral cages constructed by C_3 -symmetric building blocks and NH_4^+ node-assembled clusters, which are termed as "supramolecular secondary building units (SSBUs)", akin to the "secondary building units" (SBUs) in MOF chemistry.

We agree with the reviewer that this is NOT the first time that supramolecular entities are used to construct H-bonded porous structures. We respect the contributions made by the pioneering researchers, especially the wonderful work in Tohnai's group involving the employment of sulfonic acid and organic amine clusters for constructing porous HOFs. In the meantime, we are still confident that our study has independent novelty and significance. We particularly highlight the adaptable role of inorganic NH_4^+ node-assembled clusters in crystal engineering driven by H-bonding assembly, which deserves more attention of the research community of HOFs and broadened supramolecular chemistry.

Specifically, since inorganic NH_4^+ has a tetrahedral geometry with four N-H donors acting like metal ions, NH_4^+ node-assembled clusters are more versatile than ordinary SSBUs when participating in H-bonding assembly with different scaffolding molecules. Except for the successful assembly of the body-centered cubic HOF in this study, the

versatility of NH_4^+ nodes has also been demonstrated in our previous work in which NH_4^+ nodes regulated the assembly of C_2 -symmetric carboxylic molecules (*Small*, **2023**, 19, 2207771). Most recently, we have obtained a new class of NH_4^+ node-assembled clusters termed SSBU- NH_2 -2 with columnar structure as follows (**Unpublished results**). SSBU- NH_2 -2 has a similarity with the [4+4] clusters reported by Tohnai. However, SSBU- NH_2 -2 is a continuous and columnar four NH_4^+ and four sulfonic [4+4] cluster while the former is an isolated cluster. We believe that this would bring up fresh opportunities to construct hybrid H-bonded frameworks.

Meanwhile, previously reported [4+4] clusters constructed by sulfonic acids and amines largely depended on the use of C_4 -symmetric sulfonic derivatives and TPMA. In comparison, NH_4^+ node-assembled clusters could be adaptively assembled with C_2 -, C_3 -, and C_4 -symmetric sulfonic and carboxylic molecules. The adaptability of NH_4^+ node-assembled clusters allows the use of building blocks with different molecular symmetries, which would expand the structural diversity of HOFs.

On the other hand, although H-bonded crystalline frameworks with guest accommodation in discrete cavities have been previously investigated as inclusion compounds, the currently reported HOF with octahedral cages for haloform encapsulation is highlighted for its fast, broadband photochromic function, which has not been achieved for ordinary inclusion compounds. We demonstrate in this study, for the first time, that the cavities in H-bonded frameworks can serve as spatially-confined nanoreactors, enabling reversible photo/thermal transformation of guest molecules. This makes the HOF potentially be used in environmental sensing.

Overall, this work provides a new strategy by shaping distinctive NH_4^+ node-assembled clusters to expand the framework topology of HOFs, and a new prototype of H-bonded nanoreactors to accommodate the photochemical reactions of guest molecules for

potential applications. We believe the work will be of significant interest to researchers in the area of HOFs and porous crystalline materials for multi-domain applications.

In the revised manuscript, we labelled the NH_4^+ node-assembled clusters as “SSBU-NH₂-1”, suggesting that the supramolecular secondary building units were formed based on NH_4^+ nodes. This should prevent misleading the readers that the cluster is the first supramolecular secondary building unit. Meanwhile, relevant publications by the pioneering researchers, especially those contributed by Tohnai, have been cited at appropriate positions in the main text. The publications inserted into the reference list are as follows:

[27] Tohnai, N. et al. Well-designed supramolecular clusters comprising triphenyl-methylamine and various sulfonic acids. *Angew. Chem. Int. Ed.* **46**, 2220-2223 (2007).

[28] Oka, K. et al. Ice-like dynamics of water clusters. *J. Phys. Chem. Lett.* **15**, 267-271 (2024).

[51] Ami, T., Oka, K., Tsuchiya, K. & Tohnai, N. Porous organic salts: diversifying void structures and environments. *Angew. Chem. Int. Ed.* **61**, e202202597 (2022).

[58] Sei, H. et al. Cage-like sodalite-type porous organic salts enabling luminescent molecule's incorporation and room-temperature phosphorescence induction in air. *Small.* **19**. e2301887 (2023).

[60] Sei, H., Oka, K., & Tohnai, N. Incorporation of coronene into cage-like porous organic salts and induction of its room-temperature phosphorescence in air. *ChemNanoMat.* **9**, e202300248 (2023).

REVIEWERS' COMMENTS

Reviewer #1 (Remarks to the Author):

the authors have addressed all the questions.

Reviewer #2 (Remarks to the Author):

The author has addressed the majority of inquiries, thereby satisfying the fundamental prerequisites for acceptance. Nevertheless, during my examination of the article's crystal structure, I came across a minor issue that necessitates additional elucidation from the author.

The author concluded that CHI_3 undergoes photodissociation without sufficient evidence. On the contrary, I believe that iodomethane undergoes photopolymerization after exposure to light, producing a metastable tetramer in the confined nanoreactors. The analysis of the single crystal structure shows that the C-I bond just changed from 1.96 Å to 2.06 Å after illumination, indicating no cleavage. The author should reconsider this part of the content.

● Point-by-point response to reviewers

Reviewer #1

Comments: the authors have addressed all the questions.

Response: Thanks for the comment.

Reviewer #2

Comment: The author has addressed the majority of inquiries, thereby satisfying the fundamental prerequisites for acceptance. Nevertheless, during my examination of the article's crystal structure, I came across a minor issue that necessitates additional elucidation from the author. The author concluded that CHI_3 undergoes photodissociation without sufficient evidence. On the contrary, I believe that iodomethane undergoes photopolymerization after exposure to light, producing a metastable tetramer in the confined nanoreactors. The analysis of the single crystal structure shows that the C-I bond just changed from 1.96 Å to 2.06 Å after illumination, indicating no cleavage. The author should reconsider this part of the content.

Response: We appreciate the reviewer's positive evaluation of our revised manuscript. Regarding the photochemical transformation pathway of \$\text{CHI}_3\$, the solid-state absorption spectra of \$\text{TCA_NH}_4@\text{CHI}_3\$ (Fig 5b & 5d) and particularly the EPR spectrum with iodine free radical signals (Fig 5c) confirmed the cleavage of C-I bond and formation of metastable \$\text{I}_2\$. Additionally, the UV-vis spectrum of pure \$\text{CHI}_3\$ (Fig 5e) also proved that \$\text{CHI}_3\$ easily underwent photodissociation and generation of \$\text{I}_2\$ after illumination.

As for the single crystal analysis of C-I bond after illumination mentioned by the reviewer, actually, the carbon atoms and iodine atoms became disordered along with the increase of C-I bond and the shortening of I-I distance. Please refer to the uploaded raw data file of ' \$\text{TCA_NH}_4\text{I}_2.cif\$ '. This was also indicative of the photodissociation of \$\text{CHI}_3\$ and the generation of metastable \$\text{I}_2\$. To validate the single crystal analysis while preventing the presence of free, highly disordered carbon atoms in the unit cell, a preset value of the C-I bond distance needs to be given. Thus, the obtained C-I bond distance of 2.06 Å could not reflect the photochemical transformation of \$\text{CHI}_3\$ and it could not be used as the evidence to support the formation of tetramer.

In the revised manuscript, we have added a sentence to the 'Methods' section to explain the operation in the single-crystal analysis to prevent the occurrence of disordered and free carbon atoms.

'*The bond length of C-I bond was preset to be 2.0 Å in \$\text{TCA_NH}_4@\text{I}_2\$ using \$\text{DFIX}\$ command to avoid disordered and free carbon atoms.*' (Page 6, right column)